# Mapping of a Stripe Rust Resistance Gene *Yr72* in the Common Wheat Landraces AUS27506 and AUS27894 from the Watkins Collection

**DOI:** 10.3390/genes14111993

**Published:** 2023-10-25

**Authors:** Mumta Chhetri, Hanif Miah, Debbie Wong, Matthew Hayden, Urmil Bansal, Harbans Bariana

**Affiliations:** 1Plant Breeding Institute, School of Life and Environmental Sciences, Faculty of Science, The University of Sydney, 107 Cobbitty Road, Cobbitty, NSW 2570, Australia; mumta.chhetri@sydney.edu.au (M.C.); hanif.miah@sydney.edu.au (H.M.); 2AgriBioCentre, Department of Environment and Primary Industries, La Trobe Research and Development Park, Bundoora, VIC 3082, Australia; debbie.wong@agriculture.vic.gov.au (D.W.); matthew.hayden@agriculture.vic.gov.au (M.H.); 3School of Science, Western Sydney University, Bourke Road, Richmond, NSW 2753, Australia

**Keywords:** wheat, stripe rust, KASP marker, STS marker, marker validation

## Abstract

Stripe rust, caused by *Puccinia striiformis* f. sp. *tritici* (*Pst*), is among the major threats to global wheat production. The common wheat landraces AUS27506 and AUS27894 displayed stripe rust resistance against several commercially prevailing *Pst* pathotypes. These genotypes were crossed with a stripe-rust-susceptible landrace AUS27229 to understand the inheritance of resistance and to determine the genomic location(s) of underlying gene(s). F_3_ generations of crosses AUS27506/AUS27229 and AUS27894/AUS27229 showed monogenic segregation for stripe rust resistance under greenhouse conditions. The absence of segregation for stripe rust response among the AUS27506/AUS27894-derived F_3_ population suggested that both genotypes carry the same gene. The stripe rust resistance gene carried by AUS27506 and AUS27894 was tentatively named *YrAW4*. A bulked segregant analysis placed *YrAW4* in the long arm of chromosome 2B. The AUS27506/AUS27229 F_3_ population was enhanced to develop an F_6_ recombinant inbred line (RIL) population for detailed mapping of chromosome 2BL. DArT-based SSR, STS and SNP markers were employed to enrich the 2BL map. DArT-based STS markers *sun481* and SNP marker *IWB12294* flanked *YrAW4* proximally (1.8 cM) and distally (1.2 cM), respectively. Deletion mapping placed *sun481* in the deletion bin 2BL-5. All stripe rust resistance genes, previously located on chromosome 2BL, neither express an infection type like *YrAW4*, nor are they mapped in the deletion bin 2BL-5. Hence, *YrAW4* represented a new locus and was formally named *Yr72*. The usefulness of the markers *IWB12294* and *sun481* in marker-assisted selection was demonstrated by the amplification of alleles that are different to that linked with *Yr72* in 19 common wheat and two durum wheat cultivars.

## 1. Introduction

Stripe rust, caused by *Puccinia striiformis* f. sp. *tritici* (*Pst*), is one of the most destructive diseases of wheat worldwide [1,2]. The first stripe rust incursion in eastern Australia occurred in 1979 [3] and in western Australia in 2002 [4]. Subsequently, it caused several epidemics with devastating impacts on the agricultural industry, and an approximate yield reduction of 84% in certain areas [5]. Now, the stripe rust pathogen has a history of 44 years in Australia, and it still remains a major constraint for wheat production, with an estimated average annual loss of AUD 127 m [6].

Although more than 80 loci for stripe rust resistance have been characterized and formally named (https://wheat.pw.usda.gov/GG3/wgc; accessed on 21 July 2023), most of these belong to the race-specific category. Resistance genes that condition high level of resistance often succumb to the acquisition of virulence in pathogen populations. Wellings and McIntosh [7] summarized the evolution of virulence in the Australian *Pst* population and demonstrated the stepwise attainment in virulence for stripe rust resistance genes *YrA*, *Yr6* and *Yr7* individually and in different combinations. The other commercially important event included the detection of virulence for *Yr17* in 1999. All derivatives of the 1979 introduction (104 E137A−) were virulent on genotypes carrying *Yr3* and *Yr4*. In contrast, the 2002 introduction (134 E16A+) and all its current derivatives are avirulent on *Yr3* and *Yr4*. In addition, this group carried virulence for resistance genes *Yr8* and *Yr9* that have not been deployed widely in Australian wheat cultivars. Stripe rust resistance genes *Yr6*, *Yr7*, *Yr9*, *Yr17* and *Yr27* have been postulated in spring wheat cultivars grown worldwide. Pathotypes carrying virulence for these resistance genes have been reported in many countries, including Australia.

All-stage resistance (ASR) and adult plant resistance (APR) are two main categories of resistance to rust diseases in wheat. ASR genes protect plants throughout all growth stages, while APR genes provide post-seedling resistance. To achieve long-lasting control, it is essential to combine ASR and APR genes in a single cultivar, which is now possible through the advancement in molecular technologies. This technological advancement has enabled the efficient mapping of resistance genes and identification of closely linked markers via whole genome scanning [8] using different high-throughput platforms such as diversity arrays technology (DArT) [9,10], genotyping-by-sequencing (GBS) [8], selective genotyping (SG) [11], and SNP-genotyping arrays [12]. The development of an ordered draft sequence of the 17-gb with more than 75,000 genes positioned along 21 chromosomes of wheat [13] adds further impetus to mapping and marker development research.

To achieve durable resistance to stripe rust, the discovery and characterization of diverse sources of resistance is essential. The Watkins Collection of common wheat landraces [14] was screened [H.S. Bariana unpublished results] to identify genotypes for detailed genetic analysis. Daetwyler et al. [15] conducted genomic prediction for rust resistance among 247 common wheat landraces from the Watkins Collection and predicted the presence of some uncharacterized rust resistance loci. The stripe rust resistance genes *Yr47* (AUS28183 and AUS28187), *Yr63* (AUS27955), *Yr81* (AUS 27430) and *Yr82* (AUS27969) were mapped in chromosomes 5B, 7B, 6A and 3B, respectively, and the leaf rust resistance genes *Lr52* (AUS28183), *Lr79* (AUS26582; durum landrace) and *Lr82* (AUS27352) were mapped in chromosomes 5B, 3B and 2B, respectively (https://wheat.pw.usda.gov/GG3/wgc; accessed on 21 July 2023). Two landraces, AUS27506 and AUS27894, exhibited seedling resistance against a range of *Pst* pathotypes in Australia. This study covers the molecular mapping of stripe rust resistance carried by these genotypes and identification of resistance-linked markers for marker assisted selection.

## 2. Materials and Methods

### 2.1. Plant Material

Arthur Watkins, an English botanist, collected wheat genotypes from more than 30 locations worldwide during the 1920s to 1930s [14]. A purified version of around 830 common wheat landraces is stored at the Australian Grains GeneBank, Horsham, VIC, Australia (previously known as Australian Winter Cereal Collection, Tamworth, NSW, Australia). Seed of these accessions was kindly provided by staff of the Australian Winter Cereal Collection, Tamworth, NSW, Australia. These accessions were screened under field and greenhouse conditions to identify stripe-rust-resistant accessions. Stripe-rust-resistant accessions AUS27506 and AUS27894, collected from France and Spain, respectively, were crossed with a susceptible landrace AUS27229. F_3_ populations comprising 99 and 77 families were developed from AUS27506/AUS27229 and AUS27894/AUS27229 crosses. Only AUS27506/AUS27229 F_3_ was advanced to generate an F_6_ recombinant inbred line (RIL) population for further analysis. AUS27894 and AUS27506 were crossed to determine the allelic relationship of the gene(s) carried by these genotypes. Nineteen Australian common wheat cultivars and two durum wheats were used to validate markers closely linked with the resistance gene identified in this study. These cultivars represented all three wheat belts of Australia (eastern, southern and western).

### 2.2. Greenhouse Tests

Twenty seeds of each AUS27506/AUS27229 F_3_ line and 8 to 10 seeds of each RIL were sown in 9 cm plastic pots filled with mixture of pine bark and river sand (2:1 ratio) and fertilized with 20 g of complete fertilizer Aquasol dissolved in 10 L of water. Parental genotypes and the susceptible control Morocco were sown in each experiment. AUS27894/AUS27506 F_3_ families and AUS27506/AUS27229 F_6_ RIL population were evaluated against *Pst* pathotype 134 E16A+17+27+ (Plant Breeding Institute, Cobbitty culture no. 617). This pathotype is virulent on stripe rust resistance genes *YrA* (*Yr73*+*Yr74*), *Yr6*, *Yr7*, *Yr8*, *Yr9*, *Yr17* and *Yr27*. It is avirulent on *Yr72*. Material was kept in rust-free temperature-controlled room maintained at 20 °C after sowing and watered daily. One-week-old seedlings were fertilized with 20 g of Urea dissolved in 10 L of water. Inoculations were made at the 2-leaf stage (10–12 days after sowing) using purified *Pst* inoculum (134 E16A+17+27+) stored in liquid Nitrogen at the Plant Breeding Institute, Cobbitty, NSW, Australia. The collection and storage procedure of inoculum is described in McIntosh et al. [16]. Stripe rust inoculations were performed according to the procedure described in Randhawa et al. [17]. Inoculated material was incubated on water-filled still trays covered with polythene hoods at 9 to 12 °C for 24 h and was then moved to temperature- and irrigation-controlled rooms maintained at 17 ± 2 °C [17]. Seedling stripe rust responses were recorded using the 0 to 4 infection type (IT) scale after 14 to 16 days after inoculation [16]. The ITs 0, 1, 2 and 3 represented resistance, whereas ITs 3+ and 4 represented susceptibility. Some laboratories use 0 to 9 scale for scoring seedling stripe rust responses described in McNeale et al. [18]. Correspondence of these two scales is provided in McIntosh et al. [16].

### 2.3. Isolation of Genomic DNA

Leaf tissue was collected from 12-day-old seedlings in 2 mL Eppendorf tubes and was dried on silica gel for 72 h prior to DNA isolation. Genomic DNA was extracted using the modified CTAB method [19] and quantified using Nano Drop spectrophotometer (NanoDrop^®^ ND1000; Thermo Fisher Scientific Inc., Waltham, MA, USA). One µg DNA for each RIL, artificial F_1_ and both parents were sent for high-throughput genotyping and working dilutions of 30 ng/µL were made for in-house use.

### 2.4. Bulked Segregant Analysis

Bulked segregant analysis (BSA) was performed on resistant and susceptible bulks prepared by pooling equal amounts of DNA from 20 homozygous resistant (HR) and 20 homozygous susceptible (HS) F_3_ lines, respectively. The F_1_ artificial bulk was prepared by combining an equal amount of DNA from the rest of the population. DNA of bulks and F_1_ were sent to Diversity Arrays Technology for BSA using high-density DArT array wheat *Pst*I (*Taq*I) 3 (http://www.diversityarrays.com; accessed on 21 July 2023).

### 2.5. Sequence Tagged Site (STS) and Simple Sequence Repeat (SSR) Genotyping

Sequences of resistance-linked DArT clones (kindly provided Dr A. Kilian) were used to develop STS markers using the Primer3 program [20]. In addition, 52 SSR markers previously mapped on chromosome 2BL (http://wheat.pw.usda.gov; accessed on 21 July 2023) were also genotyped to enrich the genetic map [21]. PCRs were carried out in 10 µL reaction volume containing 2 µL of 30 ng/µL DNA, 1 µL of 1X PCR buffer containing MgCl_2_, 0.75 µL of dNTPs, 0.4 µL of forward (1.25 mM) and reverse (5.0 mM) M13 tagged primers, 0.1 µL (0.5 mM) of M13 fluorescent tag and 0.04 µL of Taq DNA polymerase. PCR amplification conditions described in Randhawa et al. [17] were used. Markers that were polymorphic on parents and showed strong linkage among HR and HS lines were evaluated on the entire AUS27506/AUS27229 RIL population. Electrophoresis of PCR products was carried out on 2.5–3% agarose (Amresco Inc., Solon, OH, USA) gel stained with GelRed™ (Biotium Inc., Fremont, CA, USA). PCR products were visualized using UV gel documentation system. Markers which could not be differentiated on agarose gel were separated on polyacrylamide gel using Analyzer Gene ReadIR 4300, Li-COR sequencing system (Li-COR Bio-sciences, Lincoln, NE, USA) after denaturing PCR product at 95 °C for five minutes.

### 2.6. SNP Genotyping

Twenty-seven SNPs showed strong linkage with the stripe rust resistance gene in AUS27506 AUS27894. Closely linked SNPs (*IWA5694*, *IWA5839*, *IWA6130*, *IWA6334*, *IWA6417*, *IWA7265*, *IWB10417*, *IWB10455*, *IWB12294*, *IWB20875*, *IWB21638*, *IWB23209*, *IWB24984*, *IWB25015*, *IWB28191*, *IWB31823*, *IWB33668*, *IWB43166*, *IWB45530*, *IWB45652*, *IWB49793*, *IWB49793*, *IWB62498*, *IWB62757*, *IWB62762*, *IWB69000* and *IWB79078*) were used to design competitive allele-specific PCR assays (KASP) with two allele-specific and one common primer. These markers were genotyped on the parents and the CFX96 Touch™ Real-Time PCR Detection System (Bio-Rad, Hercules, CA, USA) was used for comparing KASP amplifications among different genotypes following the protocol given in KASP genotyping manual (https://biosearch-cdn.azureedge.net/assetsv6/KASP-genotyping-chemistry-Use; accessed on 21 July 2023).

### 2.7. Deletion Bin Mapping of Linked Markers

The DNA samples of chromosome 2BL deletion stocks (del2BL-9, del2BL-3, del2BL-7, del2BL-5 and del2BL-6 [22]), kindly provided by Dr Evans Lagudah, were used to confirm the location of resistance gene. Chinese Spring (CS) was used as control.

### 2.8. Data Analysis

Goodness of fit of observed segregation data to the expected genetic ratios was tested through Chi-squared analysis. Linkage map was constructed using Mapmaker version 3.0 [23] and recombination fractions were transformed to centi Morgans (cM) using Kosambi mapping function [24]. Final genetic linkage map was constructed using MapChart [25].

## 3. Results

### 3.1. Inheritance Studies

The seedling stripe rust responses of AUS27506 and AUS27894 varied from 2C to 3C, when evaluated against *Pst* pathotype 134 E16A+17+27+. This pathotype was chosen due to its predominance in farmers’ fields in eastern Australia and virulence for stripe rust resistance genes *Yr17* and *Yr27*. The stripe rust screening results of F_3_ populations derived from crosses AUS27506/AUS27229 and AUS27894/AUS27229 are presented in Table 1. Both populations showed monogenic inheritance of stripe rust resistance. Both parents and homozygous resistant F_3_ lines produced similar stripe rust responses, suggesting the presence of the same gene. AUS27506 and AUS27894 were crossed, and F_3_ families were generated. All 80 F_3_ families produced responses similar to the parents, confirming the presence of the same gene in both genotypes. The resistance locus segregating in both populations was temporarily named *YrAW4*. The AUS27506/AUS27229-derived population was advanced to generate an F_6_ RIL population. The segregation for *YrAW4* among the RIL population was confirmed (Table 1).

### 3.2. Molecular Mapping

Twenty homozygous resistant and homozygous susceptible lines, each from the AUS27506/AUS27229 F_3_ population, were used to construct contrasting bulks. These bulks were subjected to BSA using DArT markers. DArT markers *wPt-2397*, *wPt-665550*, *wPt-7161*, *wPt-8916*, *wPt-1722*, *wPt-4197*, *wPt-6542*, *wPt-9104.2*, *wPt1650.2*, *wPt-2185*, *wPt-0510*, *wPt-0948*, *wPt-3632*, *wPt-7350*, *wPt-9104.1*, *wPt-9190* and *wPt-9336* showed linkage with *YrAW4*. All these markers are located in chromosome 2BL. Linked DArT clones were converted to STS markers and tested on the AUS27506/AUS27229 RIL population. The STS primers were named “sun” (Sydney University). Three STS markers (Table 2), *sun481* (*wPt-665550*), *sun482* (*wPt-7161*) and *sun483* (*wPt-2397*), showed polymorphism between parents and were genotyped on the entire RIL population. The markers *sun481* and *sun482* flanked *YrAW4* at genetic distances of 1.8 cM (proximally) and 2.7 cM (distally), respectively (Figure 1). Marker *sun483* mapped 5.8 cM distal to *sun482*.

### 3.3. Deletion Mapping of Flanking Markers

The *YrAW4*-linked STS markers *sun481* and *sun482* were genotyped on DNA of the 2BL deletion stocks del2BL-9, del2BL-3, del2BL-7, del2BL-5 and del2BL-6 to determine their precise genomic locations. Chinese Spring, AUS27506, AUS27894 (resistant parents) and AUS27229 (susceptible parent) were used as controls. The amplicons produced by deletion stocks were compared with Chinese Spring. The marker *sun482* amplified the expected product in del 2BL-5, del2BL-6 and CS, whereas *sun481* amplified the target band only in del 2BL-5 and CS. These results placed the closest marker *sun481* in the deletion bin 2BL-5 (Figure 1b,c).

### 3.4. Saturation of Chromosome 2BL Using SNP Markers

The AUS27506/AUS27229 RIL population was genotyped using 90K Infinium SNP wheat array. Twenty-one SNP markers on chromosome 2BL mapped close to *YrAW4* and spanned from the 759 to 784 Mbp region in the physical map of Chinese Spring (IWGSC RefSeq_V2.0). These were converted into KASP assays, and those showing parental polymorphisms were genotyped on the entire population (Figure 1). Eight KASP markers were incorporated into the AUS27506/AUS27229 chromosome 2BL map (Table 3, Figure 1). The final map included 1 SSR, 3 STS and 8 KASP markers (Figure 1c). The SNP marker *IWB12294* mapped 1.2 cM distal to *YrAW4* and 1.5 proximal to *sun482* (Figure 1). Sequences of the primers for markers included in the final map are given in Table 2 and Table 3.

### 3.5. Validation of YrAW4-Linked Markers

A set of 19 Australian common wheat and two durum wheat cultivars was tested with the flanking markers *IWB12294* and *sun481*. *IWB12294* produced the SNP allele “G” in the resistant parent AUS27506 and allele “T” in the susceptible parent AUS27229 (Table 4 and Figure 2). The STS marker amplified the *YrAW4*-linked *sun481*_240bp_ and *sun481*_100bp_ alleles in the resistant and susceptible parents, respectively (Table 4). Amplification of the “T” allele of the SNP marker *IWB12294* and the *sun481*_100bp_ allele of the STS marker in all test cultivars confirmed the usefulness of these markers in marker-assisted selection of *YrAW4* in breeding programs.

## 4. Discussion

This study identified a new stripe rust resistance locus in landraces AUS27506 and AUS27894. Different genomic resources were used to determine the chromosomal location of *YrAW4* and to identify closely linked markers for marker-assisted selection of this gene. The absence of segregation among the AUS27894/AUS27506 F_3_ population demonstrated the presence of the same gene in both genotypes. The resistance locus was temporarily named *YrAW4*. DArT-based BSA indicated the location of *YrAW4* in the long arm of chromosome 2B, and deletion mapping of flanking DArT markers refined the location of *YrAW4* in the deletion bin 2BL-5. Markers *sun481* and *sun482* mapped 1.8 cM proximal and 2.7 cM distal to *YrAW4*, respectively. The target genomic region was enriched by converting *YrAW4*-linked SNP markers to KASP assays. Marker *IWB12294* mapped 1.2 cM distal to *YrAW4* and 1.5 cM proximal to *sun482* (Figure 1c).

Seedling stripe rust resistance genes *Yr5*, *Yr7*, *YrSp*, *Yr43*, *Yr44* and *Yr53* were previously reported in the long arm of chromosome 2B (https://wheat.pw.usda.gov/GG3/wgc; accessed on 21 July 2023). *Yr5* and *Yr7* were mapped 21cM away from the centromere. *Yr7* was proved to be an allele of *Yr5* [27]. *Yr5* produces ITs from 0 to hypersensitive fleck (;) against the Australian *Pst* pathotype 134 E16A+ and its derivatives, and *Yr7* is not effective against this group of pathotypes; hence, *YrAW4* cannot be either of these genes. Moreover, the *Yr5*-linked marker *Yr5STS7/8* [28] segregated independently of *YrAW4* (data not presented). Digenic segregation among the *YrAW4*/*YrSp* F_2_ population (H.S. Bariana unpublished results) also excluded the possibility of *YrAW4* being *YrSp*.

*Yr43* [29], *Yr44* [29] and *Yr53* [26] produce IT 2 on a 0 to 9 scale, which is much lower than the IT 2C-3C (on a 0 to 4 scale) produced by *YrAW4.* Xu et al. [26] placed flanking markers for *Yr43*, *Yr44* and *Yr53* in the chromosomal deletion bin 2BL-3, which is next to the centromeric deletion bin 2BL-9 (Figure 1a,b). The *YrAW4*-linked marker *sun481* was mapped in the deletion bin 2BL-5. Based on infection type and deletion bin location, we believe that *YrAW4* is a unique locus, and therefore, *YrAW4* was formally named *Yr72*. Baranwal et al. [30] also located a seedling stripe rust resistance gene *YrAW12* in chromosome 2B of the Tunisian landrace AUS26670. Based on the susceptible responses of AUS27506 (*Yr72*) and AUS26670 (*YrAW12*) against the *Pst* pathotype 239 E237A-17+33+ and their location in the long arm of chromosome 2B, these genes were concluded to represent the same locus.

The usefulness of markers closely linked with the resistance gene requires validation across potential backgrounds in which the target gene has to be transferred. The presence of the resistance-linked allele in genotypes carrying the target gene is referred to as “positive validation” and the absence/amplification of an alternate allele in cultivars lacking the target locus is called “negative validation”. Since *YrAW4* has not yet been deployed in modern cultivars, “negative validation” was performed. The markers *sun481* and *IWB12294* were negatively validated among a set of Australian common wheat and durum wheat cultivars (Table 4 and Figure 2). These robust markers can be used (Table 2 and Table 3) for marker-assisted selection of *Yr72* against these backgrounds. Parental polymorphisms need to be checked prior to the use of *sun481* and *IWB12294* for marker-assisted selection of *Yr72* in breeding programs.

Markers linked with ASR genes (https://wheat.pw.usda.gov/GG3/node/657; accessed on 21 July 2023) that are effective against currently predominant *Pst* pathotypes and race-non-specific APR genes *Yr18* [31], *Yr36* [32] and *Yr46* [33] are available to deploy different combinations of these genes in future wheat cultivars. SNP markers linked with economic traits are preferred by breeding companies for their amenability for high-throughput testing. The *YrAW4*-linked SNP marker *IWB12294* offers this opportunity (Table 3) for marker-assisted selection and marker-assisted pyramiding of this gene with other ASR and APR genes. *YrAW4* is currently being transferred to Australian cultivars through marker-assisted selection.

Both AUS27506 and AUS27894 carry a high level of resistance to stripe rust against all Australian *Pst* pathotypes, except the pathotype 239 E237A-17+33+. AUS27506 was scored moderately resistant against the predominant pathotypes of *P. graminis* f. sp. *tritici* (*Pgt*) and *P*. *triticina* (*Pt*), the causal agents of stem rust and leaf rust of wheat (H.S. Bariana unpublished results). In contrast, AUS27894 produced moderately resistant and moderately susceptible responses against Australian *Pgt* and *Pt* pathotypes, respectively. Leaf rust in Aus27506 was controlled via two APR loci, one each located on chromosomes 1B and 2D [34]. Therefore, the use of AUS27506 as a donor source in wheat breeding programs will offer an opportunity for selection of triple rust resistance.

## 5. Conclusions

The presence of a single stripe rust resistance gene in the long arm of chromosome 2B of landraces AUS27506 and AUS27894 was demonstrated by using bi-parental mapping populations. Comparison of genomic locations and seedling responses of stripe rust resistance genes previously located in chromosome 2BL (*Yr5*, *Yr7*, *Yr43*, *Yr44*, *Yr53* and *YrSp*) demonstrated the uniqueness of stripe rust resistance carried by AUS27506 and AUS27894. This locus was permanently named *Yr72*. The markers *sun481* and *1WB12294* flanked *Yr72* at 1.8 cM and 1.2 cM at the proximal and distal sides. The absence of *Yr72*-linked marker alleles of both markers in 19 Australian common wheat and two durum wheat cultivars demonstrated the usefulness of these markers in marker-assisted selection/pyramiding of *Yr72* in wheat improvement programs. The marker *sun481* is a sequence-tagged site type and will be suitable for use in wheat improvement programs where on-gel-based detection systems may not be available, whereas *IWB12294* is a single nucleotide polymorphism-based marker and is suitable for high-throughput systems. The moderately resistant responses of AUS27506 to predominant Australian *Pgt* and *Pt* pathotypes would facilitate the transfer of triple rust resistance in future wheat cultivars if this accession is used as a donor source of *Yr72*.

## Figures and Tables

**Figure 1 genes-14-01993-f001:**
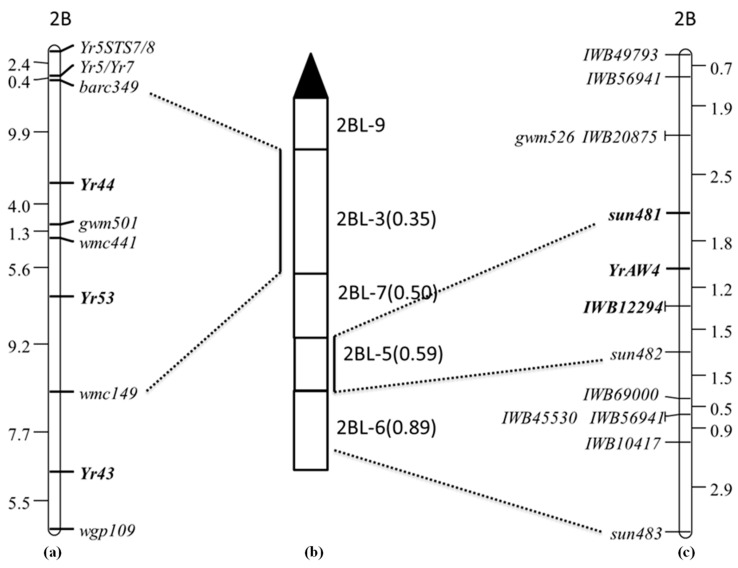
Diagrammatic representation of (**a**) stripe rust resistance genes *Yr5/Yr7*, *Yr44*, *Yr53* and *Yr43* as given in Xu et al. [26]; (**b**) deletion bins; (**c**) genetic linkage map of AUS27506/AUS27229 RIL population showing the location of *YrAW4.* Genetic distances in (**a**,**c**) are in centiMorgan (cM).

**Figure 2 genes-14-01993-f002:**
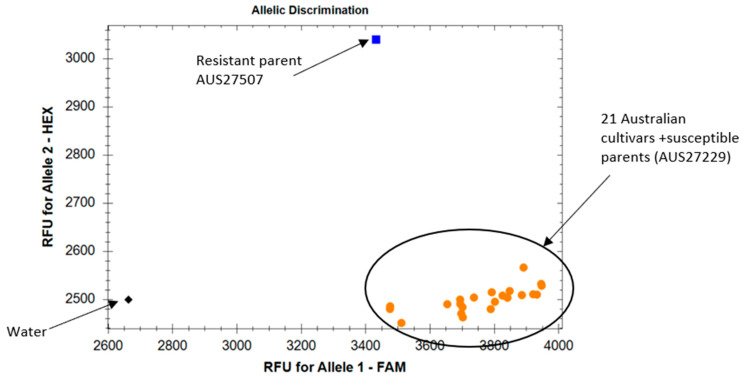
Allelic discrimination plot of SNP marker *IWB12294* using KASP assay. Allele 2 (blue) represents AUS27506 carrying *YrAW4* and Allele 1 (orange dots) represents Australian cultivars and susceptible parent AUS27229.

**Table 1 genes-14-01993-t001:** Distribution of F_3_ families and RIL lines when tested with *Pst* pathotype 134 E16A+17+27+.

Stripe Rust Response	Infection Type	Number of Lines		χ² _(1:2:1) or (1:1)_
		Observed	Expected	
F3 AUS27506/AUS27229				
Homozygous Resistant	23C to 3C	24	24.75	0.00
Segregating	23C-3C, 3+	58	49.50	1.50
Homozygous Susceptible	3+	17	24.75	2.40
Total		99	99.00	3.90 *
F3 AUS27894/AUS27229				
Homozygous Resistant	23C to 3C	19	19.25	0.00
Segregating	23C-3C, 3+	44	38.50	0.80
Homozygous Susceptible	3+	14	19.25	1.40
Total		77	77.00	2.20 *
AUS27506/AUS27229 RIL population				
Homozygous Resistant		53	51.0	0.08
Homozygous Susceptible		49	51.0	0.08
Total		102	102.0	0.16 *

* Table values of χ² at *p* = 0.05 is 3.84 and 5.99 at 1 and 2 *d.f.*

**Table 2 genes-14-01993-t002:** Primer sequences of sequence tagged site markers derived from DArT- markers spanning the *YrAW4* region.

STS Markers	DArT- Markers	Forward ^a^	Reverse
*sun481*	*wPt-665550*	GGCCAAGGTATGTTGATCGT	TCCAAATGAAACCAGGAAGG
*sun482*	*wPt-7161*	GCTCCTCTCGTTGATTGAG	GATGCAAAGGGAGAAAGCTG
*sun483*	*wPt-2397*	CAGTAGCATCCAACCCACT	GGGACGGATGATGAGACAGA

^a^ CACGACGTTGTAAAACGAC represents M13 sequence.

**Table 3 genes-14-01993-t003:** Primer sequences of kompetitive allele-specific polymerase chain reaction (KASP) markers designed from SNPs spanning the *YrAW4* region.

SNP	Allele 1 ^a^	Allele 2 ^b^	Common Primer
*IWB49793*	CCATGTTCAAGGGTCTCAATTA	CCATGTTCAAGGGTCTCAATTC	TCGCCTGATATATTCACAACCTTT
*IWB56941*	AGGAGGAATGGAAGTTTGATACT	AGGAGGAATGGAAGTTTGATACG	TGCAGAAAATGACAGCCTGA
*IWB20875*	AGAGTAGCATGCGAGTCTT	AGAGTAGCATGCGAGTCTC	CAAGCACTTTACAGGTTTCCG
*IWB12294*	TCCTCCGGCATTCCTCT	TCCTCCGGCATTCCTCG	AGCGTGCTGTACTTCGCC
*IWB69000*	TGTTCTAGCTAACCAGGCAA	CTGTTCTAGCTAACCAGGCAG	CGAGCCAGAGTCTGAGACCA
*IWB45530*	GAAACCAGCCGCAGAGTAT	TGAAACCAGCCGCAGAGTAC	GCAGTCAAGTTTTGGAACTGG
*IWB56941*	AAGGAGGAATGGAAGTTTGATACT	AAGGAGGAATGGAAGTTTGATACG	TGCAGAAAATGACAGCCTGA
*IWB10417*	TTTTCTTGTAGGAGGCAAAGATTTCAA	TTCTTGTAGGAGGCAAAGATTTCAC	TGATAATGTAAGCAGGAGTGTGCAATGTA

^a^ Allele 1 labeled with FAM: GAAGGTGACCAAGTTCATGCT; ^b^ Allele 2 labeled with HEX: GAAGGTCGGAGTCAACGGATT.

**Table 4 genes-14-01993-t004:** Validation of closely linked STS marker *sun481* and KASP marker *IWB12294* on Australian cultivars known to lack *Yr72*.

Cultivar and RIL	*sun481* (bp)	*IWB12294* (G:T)
AUS27506 and HR RIL	240	G
AUS27229 and HS RIL	100	T
Calingiri	100	T
Carnamah	250 + 120 + 100	T
Derrimut	120 + 100	T
DiamondBird	250 + 100	T
EGA Bellaroi (durum)	120 + 100	T
EGA BonnieRock	130 + 120 + 100	T
EGA Gregory	250 + 130 + 100	T
EmuRock	250 + 120 + 100	T
Espada	250 + 120	T
Forrest	250 + 120 + 100	T
Giles	255 + 100	T
Gladius	250 + 120 + 100	T
Hyperno (durum)	250 + 100	T
Magenta	250 + 120 + 100	T
Merlin	250 + 120 + 100	T
O’rion	250 + 120 + 100	T
Scout	120 + 100	T
Spitfire	250 + 120 + 100	T
Sunvale	255 + 250 + 100	T
Ventura	100	T
Wyalkatchem	250 + 100	T

## Data Availability

All data are included in this publication.

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
