# Peer review of "Mapping of a Stripe Rust Resistance Gene Yr72 in the Common Wheat Landraces AUS27506 and AUS27894 from the Watkins Collection"

_genes, 2023, doi:10.3390/genes14111993_

Round 1
Reviewer 1 Report
This paper reported a new stripe rust resistance gene Yr72 in in two common wheat landraces, and located its chromosomal position primarily. the presentation of results was rough, and need further improvement. Some tips needed to be addressed:
1. Were AUS27507 and AUS27894 from France and Spain landrace or cultivar? Were twenty-one Australian wheat cultivars resistant to stripe rust today, and the general situation of these 21 cultivars? please state clearly in Plant material section! & table 4 to provide more info. for farmers.
2. add the references for the investigation of IT in this study
3. explain the reason of the Insufficient number of markers in two parents using DArT in fig. 2
4. please list the STS and SNP markers' (primes) physical positions in reference genome sequence, add physical map in fig. 2
5. explain the reason of the 3 or 2 amplified bands of some cultivars, i.e., Carnamah Derrimut
6. Fig.1 is blurry
7. add P-value in Table 1 and the penultimate column is chi-value?
Author Response
Thanks for the suggestions and please see our response below in italic:
- Were AUS27507 and AUS27894 from France and Spain landrace or cultivar? Were twenty-one Australian wheat cultivars resistant to stripe rust today, and the general situation of these 21 cultivars? please state clearly in Plant material section! & table 4 to provide more info. for farmers.
Author response: AUS27507 and AUS27894 are pre-Green Revolution tall wheat landraces. Of these 21 Australian cultivars two were durum wheats. None of these cultivars carry Yr72. Table 4 title is modified to reflect status of Australian cultivars in relation to Yr72.
- add the references for the investigation of IT in this study
Author response: Reference added.
- explain the reason of the Insufficient number of markers in two parents using DArT in fig. 2
Author response: These are follow-up SNPs, not DArT markers.
- please list the STS and SNP markers' (primes) physical positions in reference genome sequence, add physical map in fig. 2
Author response: The STS and SNP primers are listed in Tables 2 and 3 respectively. We do not think that physical map is needed as deletion bins explain physical positions.
- explain the reason of the 3 or 2 amplified bands of some cultivars, i.e., Carnamah Derrimut
Author response: This is due to the presence of the target markers on more than homoeologue.
Figurre 1 is blurry
Author response: Figure is deleted.
- add P-value in Table 1 and the penultimate column is chi-value?
Author response: Footnote added.
Reviewer 2 Report
Chhetri et al. provided molecular markers for the stripe rust resistance gene Yr72, which will facilitate the marker-assistaed selection of resistance gene in wheat breeding. Some revision need to be considered.
1. Pst, Yr72 italic throughout the manuscript.
2. Why the authors chose only 21 Australian wheat cultivars for marker validation?
3. More details should be provided in the greenhouse infection tests (L89-94).
4. I suggest a new figure 1 with high resoulution is needed.
5. Some positive control of Yr72 also needed in Figure3.
6. the relationship between YrAW4 and YrAW12 (Baranwal et al. 2021 Molecular breeding 41:54) also need to be discussed.
Author Response
Thanks for the suggestions and please see our response below in italic:
- Pst, Yr72 italic throughout the manuscript.
Author response: Changes made.
- Why the authors chose only 21 Australian wheat cultivars for marker validation?
Author response: These cultivars represent the Northern, Southern and Western nodes of wheat belt in Australia.
- More details should be provided in the greenhouse infection tests (L89-94)
Author response: Text added.
- I suggest a new figure 1 with high resoulution is needed.
Author response: Figure 1 deleted. It is difficult to produce a new figure in a week.
- Some positive control of Yr72 also needed in Figure3.
Author response: Resistant parent is already there.
- the relationship between YrAW4 and YrAW12 (Baranwal et al. 2021 Molecular breeding 41:54) also need to be discussed.
Author response: Text added in discussion.
Reviewer 3 Report
Stripe (yellow) rust (caused by Puccinia striiformis) is a common and extremely harmful disease of bread wheat. Due to the fact that resistance genes are constantly overcome by the pathogen, the search for and mapping of new resistance genes effective against yellow rust is of priority importance. Thus, the relevance, scientific and practical significance of the work of respected authors is beyond doubt.
Thus the work of authors is definitely relevant and undoubtedly important.
The article is a high-level work and the authors obtained significant very interesting results. The article is very interesting, original, the content of the article corresponds to the abstract and title. The tables and figures are complementing the text well.
There are some comments and suggestions for authors.
1. In “materials and methods” it is necessary to describe the phytopathological part of the work in more detail. How were the plants infected? Under what conditions? How was the inoculum prepared?
Why did we use this particular fungal isolate for resistance analysis? What is its virulence formula? Also, please provide the scale used to evaluate the type of reaction (Infection
type).
2. Figure 1 is very blurry - it is better to alter it.
I believe that the authors undoubtedly obtained significant results.
Author Response
Thanks for the suggestions and please see our response below in italic:
Stripe (yellow) rust (caused by Puccinia striiformis) is a common and extremely harmful disease of bread wheat. Due to the fact that resistance genes are constantly overcome by the pathogen, the search for and mapping of new resistance genes effective against yellow rust is of priority importance. Thus, the relevance, scientific and practical significance of the work of respected authors is beyond doubt.
Thus the work of authors is definitely relevant and undoubtedly important.
The article is a high-level work and the authors obtained significant very interesting results. The article is very interesting, original, the content of the article corresponds to the abstract and title. The tables and figures are complementing the text well.
There are some comments and suggestions for authors.
- In “materials and methods” it is necessary to describe the phytopathological part of the work in more detail. How were the plants infected? Under what conditions? How was the inoculum prepared?
Author response: Text added.
Why did we use this particular fungal isolate for resistance analysis? What is its virulence formula? Also, please provide the scale used to evaluate the type of reaction (Infection
Author response: Text added.
- Figure 1 is very blurry - it is better to alter it.
Author response: Figure is deleted.
I believe that the authors undoubtedly obtained significant results.
Thanks for your kind words.
Reviewer 4 Report
Dear sir,
the document entitled 'Mapping of stripe rust resistance gene Yr72 in two common wheat landraces', seems adequate for publication in Genes. Minor flaws a mistakes are addressed in the annotated file I have attached. E.g., figure 1 are of poor quality.
I don´t fully understand who named this gene Yr72? The authors or someone else? In any case, authors should cite the reference. I would dig more into this. I would not be afraid if you were not the discoverer/s of the gene. The finding of a marker to select for it is enough and worth a publication.
Best regards

Author Response
Thanks for the suggestions and please see our response below in italic:
the document entitled 'Mapping of stripe rust resistance gene Yr72 in two common wheat landraces', seems adequate for publication in Genes. Minor flaws a mistakes are addressed in the annotated file I have attached. E.g., figure 1 are of poor quality.
I don´t fully understand who named this gene Yr72? The authors or someone else? In any case, authors should cite the reference. I would dig more into this. I would not be afraid if you were not the discoverer/s of the gene. The finding of a marker to select for it is enough and worth a publication.
Author response: Yr72 was discovered was named by our group and results were presented in M. Chhetri Ph.D. Thesis in 2015. Student moved places and some extra work was done. Eventually it was refined from thesis chapter to manuscript.
[https://wheat.pw.usda.gov/GG3/wgc; visited on July 21, 2023] – The Journal accepts this way well and hence it was not numbered.
Figure 1 deleted.
All other correction are made.